# Characterization of Endoplasmic Reticulum (ER) in Human Pluripotent Stem Cells Revealed Increased Susceptibility to Cell Death upon ER Stress

**DOI:** 10.3390/cells9051078

**Published:** 2020-04-26

**Authors:** Tae Won Ha, Ji Hun Jeong, HyeonSeok Shin, Hyun Kyu Kim, Jeong Suk Im, Byung Hoo Song, Jacob Hanna, Jae Sang Oh, Dong-Hun Woo, Jaeseok Han, Man Ryul Lee

**Affiliations:** 1Soonchunhyang Institute of Medi-bio Science (SIMS), Soon Chun Hyang University, Cheonan 31151, Korea; htw5200@gmail.com (T.W.H.); jihun@sch.ac.kr (J.H.J.); ckckck229@gmail.com (H.S.S.); hyunkyu8505@gmail.com (H.K.K.); vjvmf625@naver.com (J.S.I.); thdqudgn@naver.com (B.H.S.); 2Department of Molecular Genetics, Weizmann Institute of Science, Rehovot 761001, Israel; Jacob.hanna@weizmann.ac.il; 3Department of Neurosurgery, College of Medicine, Soonchunhyang University, Cheonan Hospital, Cheonan 31151, Korea; metatron1324@hotmail.com; 4Laboratory of Stem Cells, NEXEL Co., Ltd., 8th floor, 55 Magokdong-ro, Gangseo-gu, Seoul 07802, Korea

**Keywords:** human pluripotent stem cells, endoplasmic reticulum (ER), ER stress, binding immunoglobulin protein (BiP), C/EBP homologous protein (CHOP), proteostasis

## Abstract

Human pluripotent stem cells (hPSCs), such as embryonic stem cells (ESCs) and induced pluripotent stem cells (iPSCs), have a well-orchestrated program for differentiation and self-renewal. However, the structural features of unique proteostatic-maintaining mechanisms in hPSCs and their features, distinct from those of differentiated cells, in response to cellular stress remain unclear. We evaluated and compared the morphological features and stress response of hPSCs and fibroblasts. Compared to fibroblasts, electron microscopy showed simpler/fewer structures with fewer networks in the endoplasmic reticulum (ER) of hPSCs, as well as lower expression of ER-related genes according to meta-analysis. As hPSCs contain low levels of binding immunoglobulin protein (BiP), an ER chaperone, thapsigargin treatment sharply increased the gene expression of the unfolded protein response. Thus, hPSCs with decreased chaperone function reacted sensitively to ER stress and entered apoptosis faster than fibroblasts. Such ER stress-induced apoptotic processes were abolished by tauroursodeoxycholic acid, an ER-stress reliever. Hence, our results revealed that as PSCs have an underdeveloped structure and express fewer BiP chaperone proteins than somatic cells, they are more susceptible to ER stress-induced apoptosis in response to stress.

## 1. Introduction

Human embryonic stem cells (hESCs) originate from the inner cell mass formed during the post-fertilization blastocyst stage [1,2] Human-induced pluripotent stem cells (hiPSCs) exhibit highly similar characteristics to hESCs, and both cell types are crucial resources for identifying genetic information for early embryogenesis and organogenesis [3]. Most studies have focused on the ability of hESCs and hiPSCs to differentiate into the ectoderm, mesoderm, and endoderm, which constitute the human body, thereby suggesting their high potential as cell therapy products and sources of abundant data on early embryogenesis [4,5]. Although these studies have improved the potential for cell therapy by providing data on the transcriptomic and epigenetic modifications throughout differentiation, several questions remain unanswered. Cells that differentiate under defined conditions still show limitations in functionality and maturation, and identifying the major regulator is challenging.

In this study, we examined the differential susceptibility of hPSCs to apoptosis according to the cell organelle structure. Compared to somatic cells, hPSCs contain immature cell organelles and a specific transcriptome that determines cell fate. Particularly, mitochondria are small and round with immature cristae forms [6]. Such morphological features are known to drive adenosine tri-phosphate (ATP) production in hPSCs, which relies more on glycolysis in the cytoplasm than on the oxidative phosphorylation (OXPHOS) metabolic pathway [7]. Regarding hPSC fate conversion, the cytoplasm/nucleus ratio, changes in mitochondrial morphology, and cell membrane proteins are important indicators of stem cell fate determination. However, the morphological features of the endoplasmic reticulum (ER) in hPSCs and consequent susceptibility of stem cells have not been reported.

The ER is an organelle found in all eukaryotic cells that extends from the nuclear membrane to form a network of tubules, vesicles, and cisterna, with an overall structure resembling that of the cell membrane [8]. The ER is responsible for the proper synthesis, folding, and modification of proteins and intracellular Ca^2+^ storage [9,10,11]. Many ER proteins function as molecular chaperones to monitor the folding status of newly synthesized peptides and prevent the production of non-functional proteins as quality control machinery. When unfolded/misfolded proteins accumulate within the ER lumen, the specific unfolded protein response (UPR) signaling pathway is activated [12,13,14]. UPR signaling includes three major branches controlled by the serine/threonine-protein kinase/endoribonuclease IRE1, activating transcription factor 6 (ATF-6), or protein kinase R (PKR)-like endoplasmic reticulum kinase (PERK) [14,15,16]. Upon ER stress, PERK is released from BiP, a major chaperone protein in the ER, and is activated by autophosphorylation. Activated PERK then phosphorylates the alpha subunit of eukaryotic translation initiation factor, which attenuates global protein synthesis. Paradoxically, translation of some genes including activating transcription factor 4 (ATF4) is preferentially enhanced under this condition, which transduces the stress signal to downstream target genes including CHOP (pro-apoptotic factor) or GADD34 (eIF2a phosphatase) [14,15]. IRE1 is an ER transmembrane protein that, upon sensing folding stress in the ER, cleaves 26 internal nucleotides of the X-box-binding protein 1 (XBP-1) mRNA; the resulting spliced form of XBP-1 controls the expression of molecular chaperones and other ER biogenesis genes [17,18,19,20]. ATF6a mediates the third branch of the UPR and resides in the ER membrane. Upon ER stress, ATF6a released from BiP translocates to the Golgi apparatus, where it is cleaved by site-1 and site-2 proteases. The resulting ATF6a N-terminal fragment translocates to the nucleus to induce the expression of chaperones and enzymes essential to protein folding [21,22].

Although the role of UPR in metabolic disorders has been studied, the role of the ER structure and ER stress in hPSCs and in early human development remains unclear. In early embryogenesis, the embryo rapidly undergoes cell proliferation, and simultaneously various genes are transcribed for lineage specification [21]. The sudden changes in cell proliferation and energy metabolism result in increased intercellular levels of reactive oxygen species, and radical changes in gene transcription exceed the limit of the protein processing capacity of the ER; consequently, unfolded proteins accumulate in the ER, causing ER stress [22]. UPR is an essential event during all embryonic developmental stages and exerts beneficial effects on embryonic survival and long-term developmental potential [23,24,25]. However, if such adaptive UPR fails with prolonged ER stress, cells undergo signaling pathways leading to cell death. Hence, ER stress may have a direct role in promoting homeostasis and cell death. These findings suggest that ER stress and subsequent UPR induction is more critical to hPSCs than to fibroblasts, making hPSCs more susceptible to ER stress-mediated apoptosis. 

Therefore, we investigated the differences between hPSCs and differentiated somatic cells in terms of their response to thapsigargin (TG), an ER Ca²⁺ ATPase inhibitor [26]. We verified whether hPSCs respond to ER stress differently from fibroblasts, which were used as controls for differentiated somatic cells, with respect to apoptosis in connection with the induction patterns of CHOP and BiP. Regarding the developmental hallmark accompanying the cell fate conversion such as stem cell differentiation and somatic cell reprogramming, studies of cell organelle rearrangement and new functional acquisition are likely to provide a fundamental basis for understanding early embryogenesis and contribute to the development of stem cell therapy products.

## 2. Materials and Methods

### 2.1. Cell Culture

The hESC cell line H9 (WiCell Research Institute, Madison, WI, USA) and hiPSC cell line CMC-hiPSC-003 (Korea Centers for Disease Control and Prevention, Osong, Korea) were maintained in complete TeSR-E8 medium (StemCell Technologies, Vancouver, Canada) for feeder-free culture. These cells were cultured on vitronectin XFTM- (hPSC-qualified; StemCell Technologies) coated plates and dissociated by 3 min incubation at 37 °C with TrypLE Express (Gibco, Grand Island, NY, USA). The fibroblast MRC5 cell line was maintained in Dulbecco’s modified Eagle’s medium (DMEM high glucose; Corning, Inc., Corning, NY, USA) supplemented with 10% fetal bovine serum (PSCs quality; Corning) and 100× penicillin-streptomycin (Corning). Thapsigargin (TG, Sigma, St. Louis, MO, USA) was diluted at 10 nM to complete growth media. hESCs and hiPSCs were routinely passaged every 3 days with TrypLE Express (Gibco); fibroblasts were passaged every 4–5 days with 0.05% trypsin/ethylenediaminetetraacetic acid (EDTA). All cells were maintained in an incubator at 37 °C with 5% CO_2_. 

### 2.2. Optical Microscopy

Cells were fixed with a TEM-grade fixation solution for 60 min at 4 °C. The morphology of hESCs, iPSCs colonies, and fibroblasts was observed using an optical microscope (CKX53, Olympus, Tokyo, Japan) before sampling for transmission electron microscopy (TEM).

### 2.3. RNA Extraction, Reverse Transcription, and Quantitative Real-Time PCR

Cells were lysed with easy-BLUE (iNtRON, Daejeon, Korea) and total RNA was extracted. RNA was quantified with a Nanodrop spectrometer. One microgram of RNA was reverse-transcribed using the All-in-One 5× First Strand cDNA Synthesis Master Mix kit (CellScript, Madison, WI, USA), and quantitative PCR was performed using TOPreal™ qPCR 2X PreMIX (Enzynomics, Daejeon, Korea). The experiments were performed according to the manufactures’ protocols. RNA was extracted from each treatment condition and the experiments were performed in triplicate. Primer sequences were designed according to the Integrated DNA Technologies website (IDT, https://sg.idtdna.com/pages). All primers were validated for efficiency and the data refer to experiments performed in triplicate.

### 2.4. Western Blotting

Whole cell lysates were extracted in NP40 (Elpis Biotech, Daejeon, Korea) with a 100× protease/phosphatase inhibitor cocktail (Cell Signaling Technology, Danvers, MA, USA). The total protein concentration in each sample was measured using the Bradford assay. The protein samples were separated by electrophoresis using 12% SDS-polyacrylamide gels and transferred to 0.2-μM polyvinylidene fluoride blotting membranes (Amersham, Little Chalfont, UK). Primary antibodies were as follows: GRP 78; BiP (76-E6; 1:800; Santa Cruz Biotechnology, Dallas, TX, USA), GADD153; CHOP (B-3; 1:1 000; Santa Cruz), PARP (#9542S; 1:1 000; Cell Signaling Technology), and cleaved caspase-3 (#9661; 1:1 000; Cell Signaling Technology). The membranes were blocked for 1 h in TBS-T (50 mM Tris, 0.15M sodium chloride, 0.05% Tween 20) containing 5% blocker (BioShop, Burlington, Canada) and hybridized to primary antibody overnight at 4 °C. The membranes were washed three times for 15 min with TBS-T at room temperature and incubated for <1.5 h at room temperature with the secondary antibodies. The secondary horseradish peroxidase-conjugated antibodies were as follows: goat anti-mouse (sc-2031; 1:2 500; Santa Cruz), mouse anti-rabbit (sc-2357; 1:2 500; Santa Cruz), and goat anti-rat (sc-2006; 1:2 500; Santa Cruz). Chemiluminescence detection was performed using the Pico EPD Western Blot Detection kit (Elpis Biotech).

### 2.5. Confocal Immuno-Staining Sample Preparation

The cells were cultured on a confocal microscope dish, fixed with 4% paraformaldehyde (Biosesang, Bundang, Korea) for 20 min at 4 °C, washed twice with PBS (Corning), and incubated in 0.1% Triton X-100 (Sigma) for 5 min at 4 °C. After washing twice with PBS, the samples were blocked with 5% bovine serum albumin for 1 h at room temperature and washed with PBS. The primary antibody was incubated with the cells overnight. The dishes were washed three times with 0.05% Triton X-100 and treated with secondary antibody (1/2 000) for 2 h at room temperature in the dark. After washing with 0.05% Triton X-100, cells were counterstained with DAPI (Thermo Fisher Scientific, Waltham, MA, USA) and stored at 4 °C until confocal microscopy analysis (LSM 710; Carl Zeiss, Oberkochen, Germany). Intensity of the signals was measured with ImageJ software (NIH; Bethesda, MD, USA).

### 2.6. Terminal Deoxynucleotidyl Transferase-Mediated Deoxyuridine Tri-Phosphate (dUTP) Nick and Labeling Assay

hESCs, hiPSCs, and fibroblasts were incubated with dimethyl sulfoxide (DMSO) for 24 h and TG for 12 and 24 h, respectively. Following incubation, both suspension and adherent cells were collected and washed twice with PBS. Terminal deoxynucleotidyl transferase-mediated dUTP nick end labeling (TUNEL) staining was conducted with APO-bromodeoxyuridine (BrdU) TUNEL Assay Kit (Thermo Fisher Scientific) following the manufacturer’s instructions. Stained cells were detected by flow cytometry.

### 2.7. Library Preparation and Sequencing

Total RNA was isolated using Trizol reagent (Invitrogen, Carlsbad, CA, USA). RNA quality was assessed with the Agilent 2100 bioanalyzer using the RNA 6000 Nano Chip (Agilent Technologies, Santa Clara, CA, USA), and RNA quantification was performed spectrophotometrically (ND-2000; Thermo Fisher Scientific). Libraries were prepared from 2 µg of total RNA using the SMARTer Stranded RNA-Seq Kit (Takara Bio, Shiga, Japan). mRNA was isolated using the Poly(A) RNA Selection Kit (LEXOGEN, Inc., Vienna, Austria). The isolated mRNAs were used for cDNA synthesis and shearing following the manufacturer’s instructions. Indexing was performed using the Illumina indices 1–12. Enrichment was performed by PCR. Subsequently, libraries were checked using the Agilent 2100 bioanalyzer (DNA High Sensitivity Kit) to evaluate the mean fragment size. Quantification was performed using the library quantification kit using StepOne Real-Time PCR System (Life Technologies, Carlsbad, CA, USA). High-throughput sequencing was performed as paired-end 100 sequencing using HiSeq 2500 (Illumina, San Diego, CA, USA).

### 2.8. Gene Expression Data Analysis

To obtain the alignment file with our data (GSE130241), mRNA-seq reads were mapped using the TopHat software tool [27]. Read counts of each gene were determined based on counts obtained using Bedtools [28]. The fold-change value was obtained using DESeq2. To increase reproducibility, eight data sets comparing hPSCs to fibroblasts were retrieved from the Gene Expression Omnibus (GEO, GSE54186, ESC (H1) vs differentiated Fibroblasts; GSE54186, ESC (H9) vs differentiated Fibroblasts; GSE33298, ESC vs Fibroblasts; GSE33298, IPSC vs Fibroblasts; IPSC vs Fibroblast; GSE24487, IPSC vs Fibroblast; GSE24487, ESC vs Fibroblast; GSE20750, IPSC vs Fibroblast). Along with the RNA-seq data, these datasets were analyzed for genes showing the same fold-change direction using UpsetR. Heatmap clustering was performed with pheatmap using the ward.d2 algorithm and cutree was used to generate clusters [29]. BiNGO was used to obtain GO functional enrichments using a hypergeometric test with a false discovery rate (FDR)-corrected *p*-value cut-off of 0.001.

### 2.9. Electron Microscopy

Cells were cultured on a 100-mm culture dish, washed twice with PBS, and fixed with 2% paraformaldehyde/2.5% glutaraldehyde in 0.1M phosphate buffer (pH 7.4) for 30 min. The samples were lifted using a cell lifter (Corning), centrifuged (4000 rpm, 4 °C, 30 min) and stored at 4 °C until further processing. The samples were post-fixed in 1% osmium tetroxide, dehydrated, and embedded in Eponate-12 resin (Ted Pella, Redding, CA, USA). One-micrometer-thick section blocks were prepared using a Reichert-Jung UltraCut E ultramicrotome (Reichert Technologies, Depew, NY, USA), stained with toluidine blue, and imaged (Olympus BX-51). Seventy-nanometer-thick sections per block were placed on formvar-coated slot grids, stained with uranyl acetate/lead citrate, and imaged by TEM (H-7600, Hitachi, Tokyo, Japan) [29].

### 2.10. Flow Cytometry

We used flow cytometry (FACS Canto II, BD Biosciences, Franklin Lakes, NJ, USA) to detect the indicated fluorochromes. Positive and negative controls were obtained from the TUNEL assay kit after this assay, and samples were collected on a FACS tube. The fluorescence intensity of BrdU-Red (Ex/Em = 488/576 nm) and 7-AAD (Ex/Em = 488/655 nm) was detected and used to determine the percentage difference between each experimental group.

### 2.11. Statistical Analysis

Statistical analysis was performed using GraphPad Prism 6 software (GraphPad, Inc., San Diego, CA, USA). Data are presented as the means ± standard deviation (SD). For comparisons involving more than two groups, all analyses were performed in triplicate (at least) and statistical differences were analyzed by two-way or one-way analysis of variance with Tukey’s test; *p* < 0.05 defined statistical significance.

## 3. Results

### 3.1. Morphological Differences in ER between hPSCs and Somatic Cells 

To understand whether hPSCs and adult somatic cells respond differently to ER stress, we examined the structural features of hPSCs and fibroblasts, which were used as control cells representing adult somatic cells. All cells presented typical morphology (Figure 1A). The expression of hPSC-specific markers, OCT4 and NANOG, was verified in hESCs, hiPSCs, and fibroblasts. Both hESCs and iPSCs, but not fibroblasts, clearly expressed these markers (Figure 1B). We then evaluated the ER ultrastructure in hESCs and iPSCs by TEM to determine any features unique to the ER of these cells. These images clearly revealed less ER in hESCs and iPSCs than in fibroblasts, and the ER morphology in hPSCs was distinct from that in fibroblasts. The ER in hESCs and iPSCs had a simpler overall architecture without interconnected networks of the flat sheets in contrast to fibroblasts, which typically feature branched tube-like structures (cisternae) (Figure 1C,D). The ER is typically composed of a continuous membrane system that includes the nuclear envelope; however, the ER in hPSCs was located predominantly near the periphery of the cell. Additionally, the typical morphological features of hPSCs included a larger nucleus and smaller mitochondria compared to those in fibroblasts. Similar to the differences in nuclei and mitochondria, the ER structure was relatively underdeveloped in hPSCs compared to in fibroblasts.

### 3.2. Expression Profiles of ER-Related Genes in hPSCs Differ from Those in Fibroblasts

We investigated the expression profiles of genes functionally important for the ER of hPSCs and iPSCs and compared them with those of fibroblasts. The entire mRNA collections (transcriptome) from the hESCs, hiPSCs, and fibroblasts were sequenced (GSE130241) and the results were analyzed along with eight datasets acquired from the five different databases in the GEO [3,30,31,32,33]. We hypothesized that the structure of the ER was less developed because of downregulation of numerous genes associated with cellular organelles; we analyzed genes that were upregulated compared to in fibroblasts; however, commonly upregulated genes were clustered and were found to be unrelated to ‘ER’ or ‘ER-stress’ in the gene network. Thus, we compared downregulated genes in hPSCs. To obtain higher reproducibility and accurate results, we selected the 1929 genes commonly downregulated in hPSCs in at least six datasets (black dots in Figure 2A). The bar graphs above each dotted column indicate the number of genes that were commonly expressed. Clustering of the genes according to their fold-change pattern resulted in four clusters (Appendix A). Clusters i and ii were mainly composed of genes showing greater downregulation in hPSCs with an average log_2_FC value of 3.55 and 4.68, respectively (Appendix A). Highly enriched functions from cluster i and cluster ii were associated with cell development, e.g., skeletal system and matrix extracellular region (Appendix A). Cluster iv showed an average log_2_FC value of 2.1 and cluster iii showed the lowest average log_2_FC value of 1.13 (Figure 2B and Appendix A). Unlike the functional enrichment found in cluster iv, which consisted of general and upper-leveled ontologies, a large portion of the enrichments found in cluster iii consisted of ER and Golgi apparatus parts (Figure 2C, Appendix A). Moreover, each dataset variation in cluster iii was the lowest of all clusters, indicating that ER-related networks are commonly downregulated in most hPSC datasets with low variation (Figure 2D). Furthermore, the wide shape of the violin plot shows that this change occurred in a reproducible manner despite the batch effect from several datasets. Overall, the results of TEM and gene expression analysis suggest that the ER in hPSCs is structurally and functionally different from that in differentiated cells such as fibroblasts.

### 3.3. Embryonic Stem Cells Are Relatively Vulnerable to ER Stress, Resulting in More Severe Damage Compared to in Fibroblasts

If the ER in human embryonic stem cells remains underdeveloped, hPSCs may be vulnerable to ER stress because of the lack of homeostatic defense mechanisms. Therefore, we induced ER stress in hESCs and iPSCs with TG, which triggers depletion of the ER luminal calcium, and evaluated the potential damage by light microscopy. We observed much more intense changes in hESCs and iPSCs at the cellular level, with the colony forms reorganizing into dense cell aggregates often found detached from the culture dishes. The center of the aggregates contained many dead cells (data not shown). In contrast, fibroblasts displayed milder changes in response to the same stress conditions (Figure 3A). As the colony forms of embryonic stem cells are important for early embryonic development, TG treatment may have caused more severe damage in hESCs and iPSCs than in fibroblasts. 

The ER was observed by TEM to evaluate the morphological/structural changes resulting from 24-h TG treatment. We observed many swelled and fragmented ER structures in the fibroblasts, suggesting that these cells were experiencing ER stress (Figure 3B). Although there were underdeveloped ER in hESCs and iPSCs than in fibroblasts, the ER structures in hPSCs and fibroblasts exhibited comparable dilation and fragmentation (Figure 3B). However, hPSCs displayed different susceptibility to TG-induced ER stress because hPSCs showed different collapse of the morphological features compared to fibroblasts despite the similar structural ER stress experienced by both cell types. Thus, structural differences in cellular organelles were observed according to the status of each cell fate, indicating that the cells show different immediate responses to stress.

### 3.4. Apoptotic Response Predominated in ESCs upon ER Stress

Disturbances in the normal functions of the ER lead to an evolutionarily conserved cell stress response as compensation for damage. However, if the ER stress is too severe or prolonged, it eventually triggers cell death. Based on this, we tested whether hESCs and iPSCs were prone to apoptosis upon ER stress. TUNEL, a method used to identify and quantify apoptotic cells by detecting apoptotic DNA fragmentation, was used to identify the TG-induced apoptotic hESCs and iPSCs by labelling the fragmented DNA with GFP-dUTP. The apoptotic cell populations were greatly increased in hESCs and iPSCs following 12- and 24-h TG treatments in a time-dependent manner, in contrast to fibroblasts, which showed small indications of apoptosis under the same conditions (Figure 4A). Fluorescent cells were quantified by FACS (Figure 4B). Consistent with the histochemical results, hESCs and iPSCs were highly vulnerable to TG treatment, with increased apoptotic cell populations, compared to fibroblasts. Caspases, a family of cysteine proteases important for initiating apoptotic signal pathways, are activated by chronic ER stress [16,34]; a transient burst in poly-ADP ribosylation of nuclear proteins is also required for initiating apoptosis, followed by cleavage of poly-ADP-ribose polymerase (PARP) by caspase-3. Thus, we verified whether TG-induced apoptosis in hESCs and iPSCs was accompanied by induction of cleaved PARP and caspase-3 by Western blot analysis. We observed pronounced induction of cleaved PARP and caspase-3 in TG-treated hESCs and iPSCs, but such induction was not noticeable in fibroblasts (Figure 4C). 

To ensure that TG-induced apoptosis was mediated by the ER, we treated the hESCs and iPSCs with tauroursodeoxycholic acid (TUDCA), an effective ER stress-relieving drug. TUDCA almost completely abolished the TG-induced apoptotic cell population in hESCs and iPSCs (Figure 4D), suggesting that TG-induced apoptosis was mediated by the ER. Further, the protein expression levels of the BiP, CHOP, cleaved-PARP, cleaved-caspase 3 were distinctly decreased (Figure 4E). Thus, hPSCs entered apoptosis significantly faster than fibroblasts because of the ER stress caused by the ER stress inducer TG. In addition, the resistance capacity of hPSCs to ER stress was lower than that of fibroblasts, i.e., hPSCs were more susceptible to apoptosis.

### 3.5. hESCs and iPSCs Displayed Abundant Induction of CHOP and Lacked BiP

Because hESCs and iPSCs were prone to apoptotic responses to ER stress, we investigated whether any known ER-related pro- and anti-apoptotic factors were correspondingly expressed in TG-treated hESCs and iPSCs. We first considered the C/EBP homologous protein (CHOP or GADD153) because it is involved in ER stress-induced apoptosis [35]. Therefore, considering the high lethality of TG treatment to hESC and iPSC, we examined whether the pro-apoptotic CHOP was overly induced in hESCs and iPSCs under stress conditions. Our reverse transcription-quantitative polymersase chain reaction (RT-qPCR) data indicated that minimal CHOP mRNA was present in the three cells under non-stress conditions. However, hESCs and iPSCs induced CHOP mRNA excessively, with higher levels than those observed in fibroblasts, in response to ER stress (Figure 5A). This marked induction of CHOP in hESCs and iPSCs was confirmed at the protein level by western blotting, but such induction was not detectable in fibroblasts at the protein level even after treatment (Figure 4C). The CHOP promoter contains binding sites for many transcription factors, including ATF4, ATF6, and XBP-1, which play causative roles in CHOP transcription induction. Therefore, we investigated whether these transcription factors were overly expressed in hESCs and iPSCs in response to TG treatment. CHOP is known to directly activate GADD34, which promotes ER client protein biosynthesis in stressed cells; thus, GADD34 was included in our assays. RT-qPCR data indicated that, compared to fibroblasts, all gene transcripts were induced in hESCs and iPSCs at much higher levels (Figure 5A). 

In contrast to CHOP, BiP acts as a survival factor in ER-stressed cells by chaperoning and reducing misfolded/unfolded proteins. Blocking BiP expression promotes cell death pathways upon ER stress by increasing the tolerance to life-threatening stress [36,37]. In contrast, BiP overexpression promotes cell survival in response to the calcium ionophore [38]. We found that BiP expression was extremely low in hESCs and iPSCs under non-stress conditions compared to that in fibroblasts (Figure 5B). Given the role of BiP in initiating the UPR signaling pathway, low amounts of the BiP protein trigger stem cells to activate the UPR more rapidly and strongly. Additionally, such low levels of BiP may result in a decreased protective effect in stem cells upon ER stress, making them more susceptible to ER stress. 

Collectively, our results show that hESCs were clearly different from somatic cells, presenting higher vulnerability when challenged by ER stress. This may be because of the unique ability of ESCs to up- and downregulate the expression of the two key ER-associated pro-apoptotic and pro-survival proteins, CHOP and BiP, respectively. It remains to be determined whether the preferential apoptotic cell death under ER stress conditions detected in the ESCs is as common as that in adult fibroblasts. When treated with the same concentration of TG, hPSCs express UPR genes faster than fibroblasts, and differences in the baseline expression of ER stress sensor proteins determine the sensitivity of UPR response. This can be attributed to the fact that hPSCs, with their inherent cellular characteristics, have underdeveloped organelles and relatively low expression of relevant genes, so that intracellular stress would cause a more rapid increase of UPR gene expression in an attempt to maintain cellular homeostasis.

## 4. Discussion

hESCs originate from the inner cell mass of the early-stage preimplantation embryo and, consequently, represent the only human cell line that allows in-depth study of early human embryogenesis [1]. Pluripotency, self-renewal, and cell fate decisions of hESCs involve unique regulatory factors and mechanisms that are distinct from those of somatic cells and that precisely control the embryogenesis process [39,40]. For this, the ER and protein quality control mechanisms are the most attractive candidates for understanding the pluripotency, differentiation, and apoptosis of hESCs.

For hPSCs, including hESCs and hiPSCs, to maintain pluripotency while undergoing self-renewal, a set of highly developed mechanisms is required to protect cells from genome mutation and for cytoplasmic quality control. The mechanism underlying DNA damage and repair has been discussed elsewhere [41,42]. Here, the morphology of cell organelles and mechanism of apoptosis were examined to identify the different mechanisms in hPSCs for maintaining ER stress-associated proteostasis compared to in somatic cells. The ER morphological features related to protein synthesis, folding, and secretion, are the key determinants of proteostasis [43,44] and are regarded as highly important because when a problem occurs in proteostasis, the proteins involved in hPSC pluripotency and self-renewal may be damaged. 

We compared hPSCs and fibroblasts and found that hPSCs had a less developed ER structure and showed overall decreased expression of ER structure- and UPR-related genes. Thus, hPSCs may rapidly increase their UPR gene expression upon a sudden ER stress-inducing condition (e.g., TG treatment) to reduce ER stress. Particularly, the protein expression of BiP, an important factor that initiates the UPR signaling pathway upon ER stress, was relatively lower in hPSCs. BiP binds to PERK, IRE1, and ATF6 to inhibit signaling under normal conditions. When unfolded proteins accumulate in the ER or alterations in Ca^2+^ concentrations occur, BiP is released from the proteins, initiating the UPR. Increasing evidence suggests that increased BiP expression protects cells from ER stress-mediated cell death with decreased UPR induction [45,46]. In contrast, reduced BiP levels make the cells more susceptible to ER stress with increased UPR induction [47,48]. As hPSCs displayed an underdeveloped ER structure and low BiP protein levels and ratio over the UPR sensors, they had a relatively weaker ability to regulate ER stress upon exposure, which led to rapid increases in the expression of UPR genes (Figure 5B and Appendix A). In addition, hESC and hiPSC are derived from different origins, but these two cells share similar ER structures and response patterns to TG treatment.

When undifferentiated hPSCs undergo differentiation, changes in the transcriptome in the nucleus and maturation or reconstruction of intracellular organelles should occur simultaneously to enable the cells to differentiate into more complete functional somatic cells. For example, beta cells, which synthesize and secrete insulin, should have well-developed ER structures. Here, the mature organelles, particularly the ER, create a more sophisticated sheet structure to acquire the adequate capacity for protein synthesis and secretion. 

Another interesting possibility is that undifferentiated hPSCs promote relatively rapid signaling towards death when damaged (unfolded or misfolded) proteins accumulate in the ER. hPSCs retain chaperoning activity at sufficient levels for maintaining proteostasis in undifferentiated culture conditions. However, an underdeveloped ER structure and low amount of BiP protein in hPSCs make them highly susceptible to even a small change in the proteostasis network. As BiP has a higher affinity to cause the formation of unfolded/misfolded proteins, a small increase in the number of dysfunctional proteins can cause dissociation of BiP from the UPR sensor proteins, which generally triggers UPR signaling quickly and more robustly. Given that BiP sequesters and represses UPR signaling under normal conditions, lower BiP amounts may explain why UPR induction occurs relatively earlier and at higher levels in hPSCs than in fibroblasts. Moreover, because the original function of BiP is to chaperone misfolded/unfolded proteins, relatively small amounts of BiP likely limit the chaperoning activity of the cells, exacerbating the disturbed proteostasis in hPSCs. CHOP is rapidly and strongly induced by the accumulation of unfolded proteins and causes apoptosis of hPSCs. The specific cellular response mechanisms of hPSCs appear advantageous for maintaining their integrity. When the BiP level was artificially increased in the liver, greater protective effect against stress were observed [49], suggesting that the levels of the chaperone protein are a limiting factor in the susceptibility of cells to stress. Interestingly, the reduction of ER stress induced after treatment with a chemical chaperone observed here significantly protected hPSCs from ER stress. 

Compared to fibroblasts, hPSCs have an underdeveloped ER structure that lowers the threshold leading to apoptosis. These mechanisms are thought to control apoptosis by responding to cellular stimuli in a more sensitive manner so that the early-stage embryo can develop correctly. In addition to the ER stress and stem cell apoptosis mechanisms discussed here, others involving autophagy, ubiquitin, and chaperone have been found in stem cells for maintaining proteostasis. The post-transcriptional regulation and translation in hPSCs and proteostasis maintenance research fields require further analysis to reveal the complex mechanisms behind proteostasis in hPSCs and identify a novel network that may delay the pathogenesis of aging and degenerative diseases.

## 5. Conclusions

Our study demonstrates that (a) hPSCs have a structurally underdeveloped ER, and the expression of the chaperone gene BiP is relatively low; (b) when hPSCs and somatic cells are exposed to the same ER stress conditions, the former cannot control these stimuli properly, thus sharply increasing UPR genes and rapidly inducing apoptosis through CHOP; and (c) apoptosis can be improved by enhancing the ER chaperone function and ER capacity through chemical chaperone treatment. Our results demonstrated how PSCs respond to stress beyond a threshold. 

## Figures and Tables

**Figure 1 cells-09-01078-f001:**
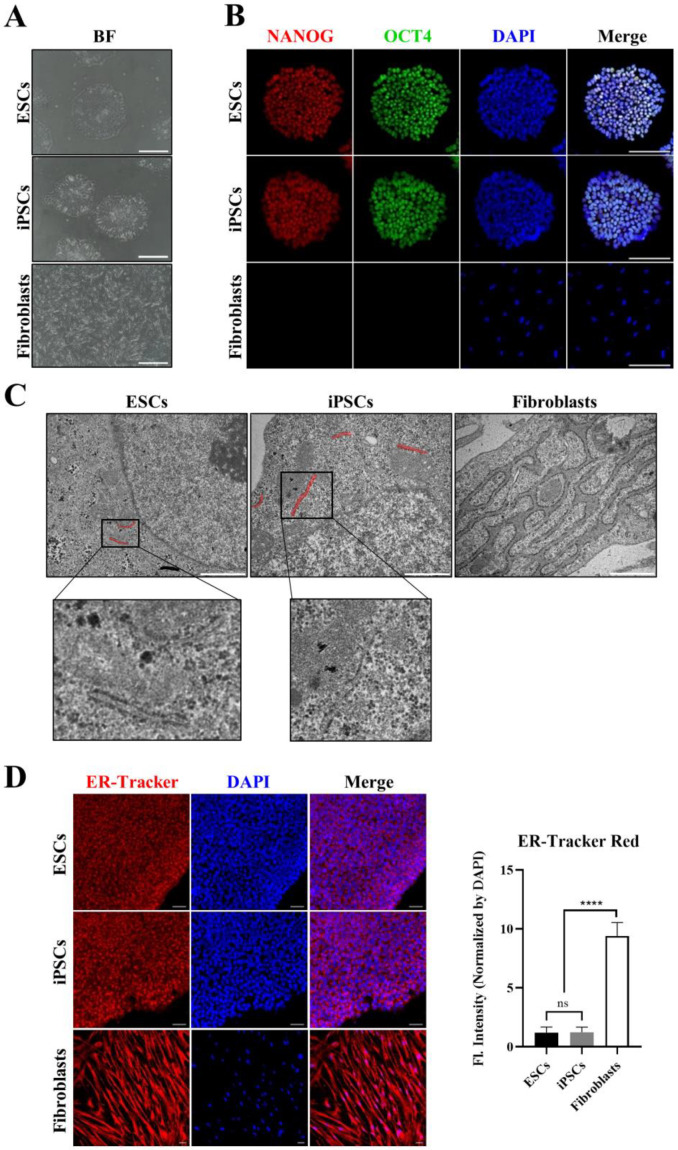
Morphological differences of the endoplasmic reticulum (ER) in human pluripotent stem cells (hPSCs) and fibroblasts. (**A**) Bright field (BF) image showing cultured cells before harvesting for transmission electron microscopy (TEM) analysis. Scale bar: 200 μm (**B**) Immunostaining showing the expression of NANOG (red fluorescence) and octamer-binding transcription factor 4 (OCT4 as known POU5F1) (green fluorescence) in hPSCs. Scale bar: 150 μm (**C**) TEM images showing the ER structure of each sample. Red bar indicates the ER in hPSCs. The small images show enlarged ER. Scale bar: 1 μm. (**D**) Confocal microscopy images showing the ER tracker signal that binds to ER. Fluorescence intensity graph indicate that 4′,6-diamidino-2-phenylindole (DAPI) normalized ER tracker signal. Scale bar: 50 μm. Values represent the mean ± SD. Tukey’s *t* test: **** *p* < 0.00001; “ns” represents non-significant differences.

**Figure 2 cells-09-01078-f002:**
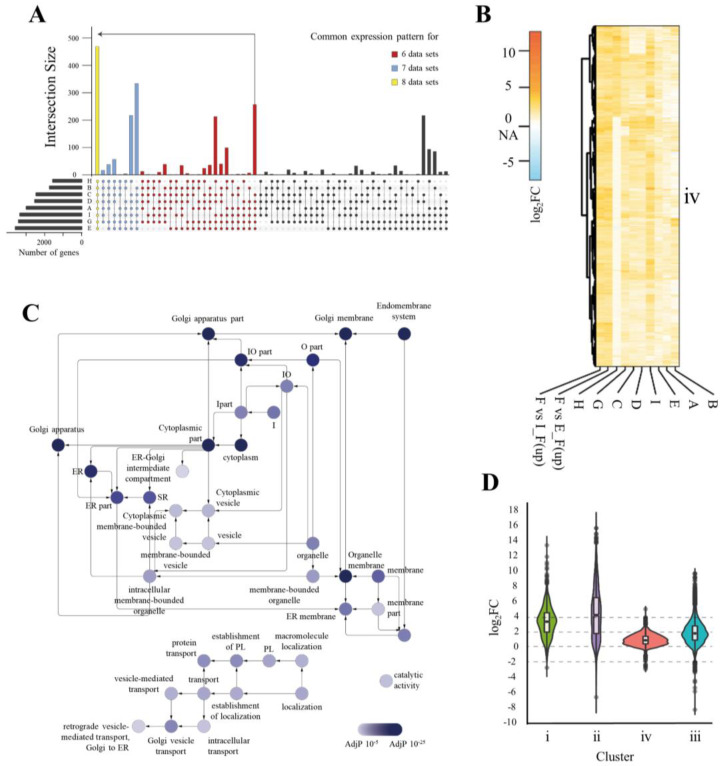
Gene expression analysis of hPSCs and fibroblasts. (**A**) UpsetR plot showing downregulated genes in hPSCs in eight datasets. The left bar graph shows the number of genes used in the intersection for each dataset. The color-filled plots show the number of commonly downregulated genes. Genes commonly downregulated in six, seven, and eight datasets are colored with red, light blue and yellow, respectively. The GEO data used are abbreviated as follows; A: GSE54186, ESC (H1) vs differentiated fibroblasts; B: GSE54186, ESC (H9) vs differentiated fibroblasts; C: GSE33298, ESC vs fibroblasts; D: GSE33298, induced pluripotent stem cells (iPSC) vs fibroblasts; E: IPSC vs fibroblasts; G: GSE24487, IPSC vs fibroblasts; H: GSE24487, ESC vs fibroblasts; I: GSE20750, IPSC vs fibroblasts; (**B**) Heat-map showing the log_2_FC-fold change value for the four generated clusters. The orange color key indicates the fold-changes of downregulated genes in hPSCs and blue color indicates the fold-changes of upregulated genes in fibroblasts. (**C**) Gene ontology (GO) network of cluster iii that satisfies a cutoff of FDR < 0.00001 for hypergeometric test with FDR correction using BiNGO which is a Java-based tool to determine GO. The darker blue-colored mapping indicates stronger enrichment. Abbreviations: endoplasmic reticulum: ER; intracellular: I; organelle: O; sub-synaptic reticulum: SR; protein localization: PL. (**D**) Violin plot showing the fold change pattern distribution of each cluster generated from the heat-map.

**Figure 3 cells-09-01078-f003:**
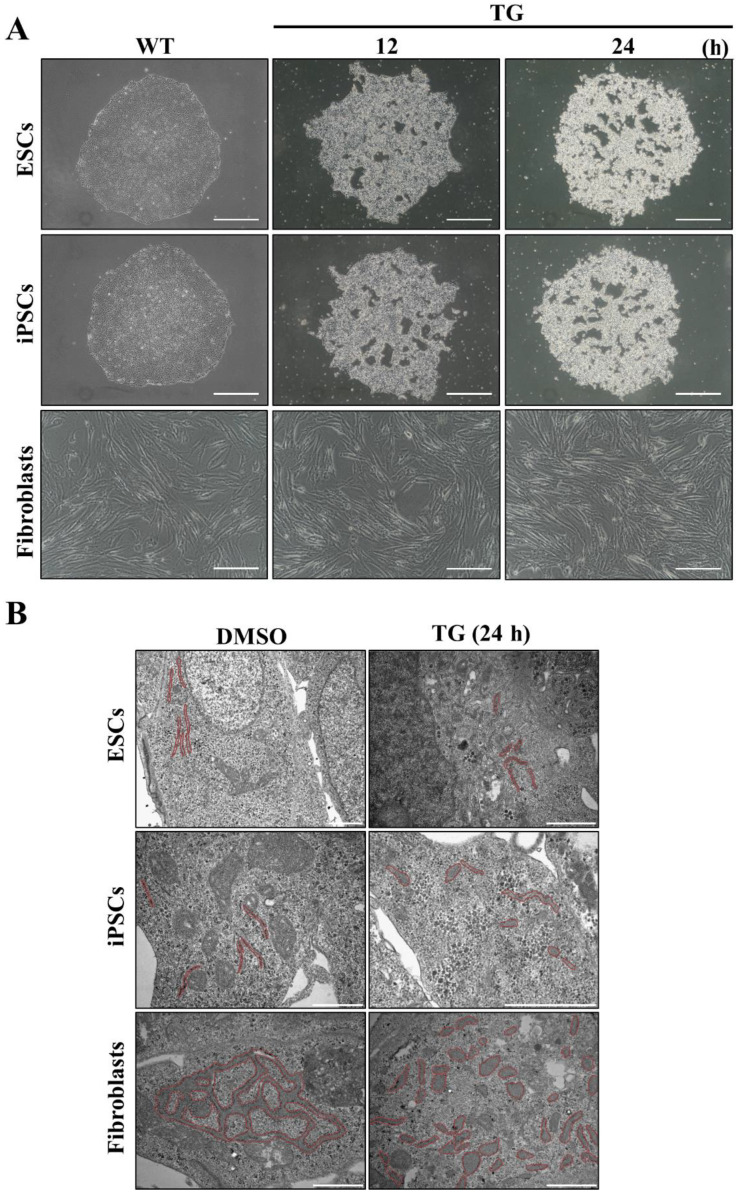
Unfolded protein response (UPR) induction under ER stress caused by thapsigargin (TG) treatment. (**A**) Morphological changes in the cell population in response to ER stress observed using an optical microscope. (WT, TG treatment for 12 or 24 h). Scale bar: 200 μm (**B**) Morphological changes in the ER structure under dimethyl sulfoxide (DMSO) or TG treatment conditions; Red line indicates the ER structure in hPSCs. Scale bar: 1 μm.

**Figure 4 cells-09-01078-f004:**
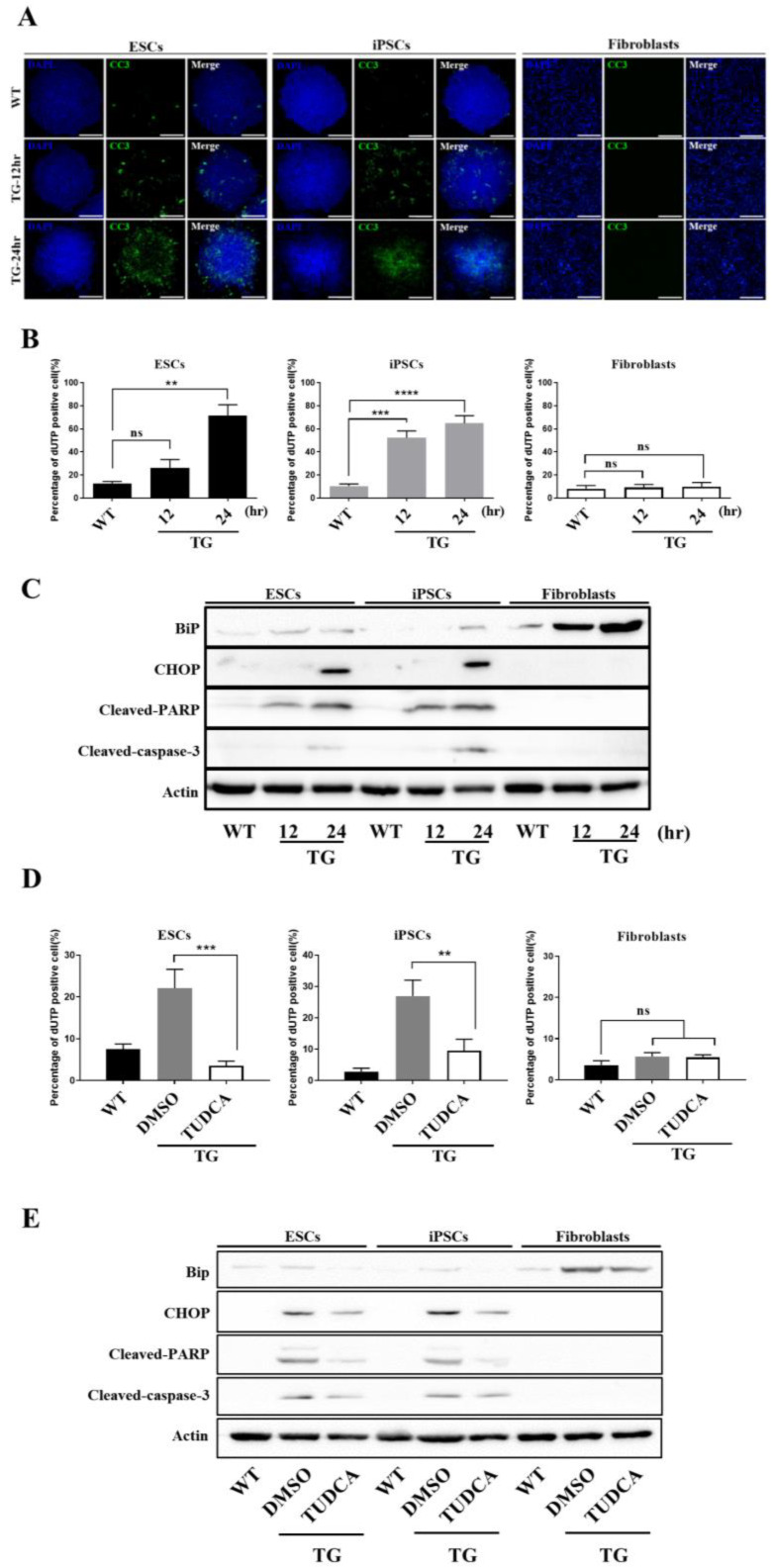
hPSCs are more susceptible to apoptosis. (**A**) Immunostaining data show cleaved caspase-3 expression by GFP fluorescence. GFP was only visible in hPSCs. Scale bar: 300 μm (**B**) Percentage of apoptosis as indicated by bromodeoxyuridine (BrdU) staining using fluorescence-activated cell sorting (FACS). Values represent the mean ± SD. Tukey’s *t*-test: ** *p* < 0.01; *** *p* < 0.001; **** *p* < 0.00001; “ns” represents non-significant differences. (**C**) Expression levels of BiP, C/EBP homologous protein (CHOP), cleaved-poly (ADP-ribose) polymerase (PARP), cleaved-caspase 3, and actin in hESCs, iPSCs, and fibroblasts treated with TG were assessed by western blotting using specific antibodies (*n* = 3). (**D**) Percentage of apoptotic treated and untreated cell lines measured by BrdU staining. Values represent the mean ± SD. Tukey *t*-test: ** *p* < 0.01 or ** *p* < 0.001. (**E**) Expression levels of BiP, CHOP, cleaved-PARP, cleaved-caspase3, and actin in hESCs, iPSCs, and fibroblasts treated with TG for 12 h and then DMSO and tauroursodeoxycholic acid (TUDCA) for 12 h were assessed by Western blotting.

**Figure 5 cells-09-01078-f005:**
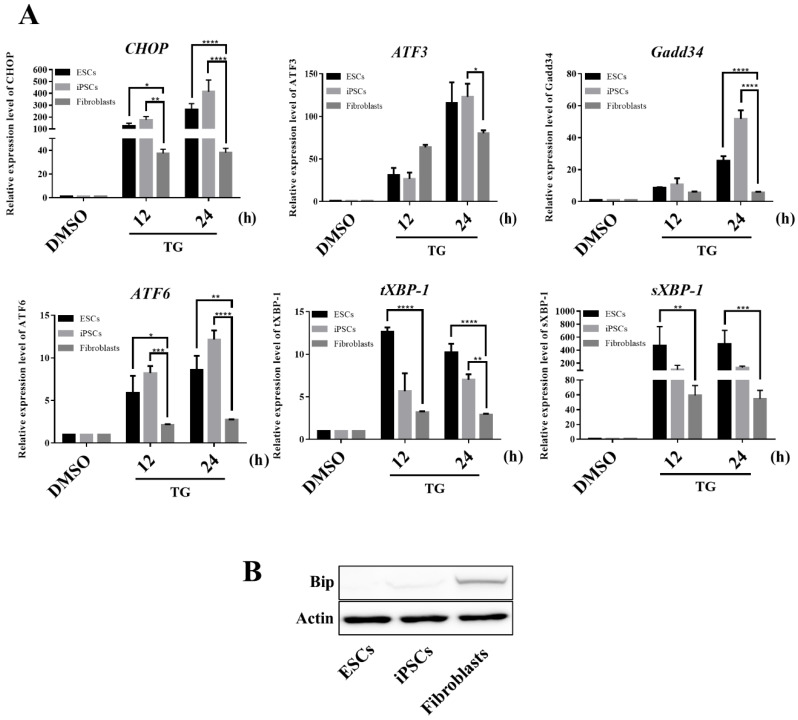
UPR induction is more significantly activated in hPSCs. (**A**) Expression levels of ER stress-associated genes in hESCs, hiPSCs, and fibroblasts during the 12- and the 24-h TG treatments. Values represent the mean ± SD. Tukey’s *t*-test: * *p* < 0.05; ** *p* < 0.01; *** *p* < 0.001; or **** *p* < 0.00001. (**B**) Expression levels of BiP and actin in ESCs, iPSCs, and fibroblasts were assessed by western blotting using specific antibodies (*n* = 3).

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
