# Peer review of "Characterization of Endoplasmic Reticulum (ER) in Human Pluripotent Stem Cells Revealed Increased Susceptibility to Cell Death upon ER Stress"

_cells, 2020, doi:10.3390/cells9051078_

Round 1
Reviewer 1 Report
This could be an interesting report, although rather descriptive, about differences between hPSCs and fibroblasts in ER morphology, sensitivity to ER stress and UPR activation. However, I have some important concerns:
-The main concern is a contradiction between the results that show more sensitivity to ER stress, which appears to cause toxicity in hPSCs and the conclusion of the authors, as indicated in the Abstract that “These results suggest a mechanism that possibly protects cells from stress in early embryogenesis”.
Upon initial stages of differentiation, protein synthesis is high and this is a natural cause of ER stress. The authors show that hPSCs have a lower capacity to adapt to ER stress and therefore instead of survival mechanisms, an apoptotic program is triggered, which is contrary to what happens in embryogenesis. This contradiction must be solved experimentally or interpreted correctly for the manuscript to be acceptable.
-Along the manuscript there is reference to a less mature or immature ER in hPSCs. What are the criteria for the maturity of the ER? Perhaps the authors should refer to smaller or underdeveloped ER?
-Fig. 3A: The differences are unclear, must be shown together with an untreated control and the differences pointed out.
-L.292 “…the ER structures in hPSCs and in fibroblasts featured comparable dilation and fragmentation (Fig. 3B).”
The effect of TG treatment seems clear in the fibroblasts, but not in the hPSCs. If there are differences with the DMSO treated cells, these should be pointed out with arrows. Perhaps an experiment staining for ER or UPR markers would make the observation more clear.
-A very low concentration of TG (10 nM as mentioned in Methods) is used. It should be commented why. This low concentration is probably the reason for lack of apoptosis in fibroblasts.
-L. 344 (Fig. 3D). There is no Fig. 3D.
-Lines 345-351. “The CHOP promoter contains binding sites for many transcription factors, including ATF4, ATF6, and XBP-1, and these play causative roles in CHOP transcription induction. Therefore, we investigated whether these transcription factors were overly expressed in hESCs and iPSCs in response to TG treatment. CHOP is known to directly activate GADD34, which promotes ER client protein biosynthesis in stressed cells; thus, GADD34 was included in our assays. RT-qPCR data indicated that, compared to fibroblasts, all the gene transcripts were induced in hESCs and iPSCs at much higher levels (Fig. 5A).”
The increase in these factors could also be a survival response, for upregulation of protective genes.
-L.397-403. “Particularly, the BiP protein expression, an important factor that initiates the UPR signaling pathway upon ER stress, was relatively lower in hPSCs. BiP is bound to PERK, IRE1, and ATF6 to inhibit signaling in normal conditions… …However, since hPSCs displayed immature ER structure and low BiP protein amounts, they had a relatively weaker ability to regulate ER stress upon exposure, which led to rapid increases in the expression of UPR genes.”
The ratio of the UPR sensors to BiP should be measured. Perhaps there is just a smaller ER, therefore less BiP, but also less PERK, IRE1, and ATF6, so the ratio would be the same.
-L.404. Paragraph starting “Although the present study described the characteristics of hPSCs by inducing ER stress through TG treatment, these findings are useful to explain the cellular responses in hPSCs in the initial stages of differentiation.”
As explained above, during initial stages of differentiation there is high adaptation to protein load, contrary to what is seen in this study.
Reviewer 2 Report
The submitted manuscript by Ha et al entitled “Characterization of Endoplasmic Reticulum in Human Pluripotent Stem Cells Revealed Increased Susceptibility to Cell Death upon ER Stress” describes a mechanism that protects cells from ER stress in early embryogenesis. Compared to fibroblast, PSC has very unique characteristics in terms of cell metabolism, cell death, and cell organelles. Researching the unique characteristics of PSC in order to develop PSC based stem cell therapy products and comparing PSC with fibroblast are very important. Ha et al. compared PSC with fibroblast in terms of the structural characteristics of ER and revealed in their manuscript how differently the difference of the ER structure reacts to the external ER-stress stimulation. Although the topic of this manuscript is timely and should be of general interest to readers of this journal, it is necessary to resolve some concerns before the manuscript is published.
The authors claim that the TG-induced apoptosis is reduced by Tudca. To prove the argument, it is required to show that ER-stress associated genes are lessened by Tudca. At least, it is necessary to show that Tudca controls the revelation of Bip and chop.
That compared to fibroblasts, PSC has differentiation potency is very distinct. Therefore, it is better to mention if these cells’ ER-stress changes in differentiation.
hESC and iPSC are representative PSC cells. In this manuscript, hESC and iPSC seem to have no difference in terms of ER-stress. It is necessary to explain that such no difference is the unique characteristics of each cell.
In Figure 2, authors mention only the reduced genes in PSC, compared to fibroblast. It is necessary to mention if the increased genes in PSC include any genetic cluster related to ER structure/stress.
In Figure 1.C and Figure 3.B, there is highlight display in order to show the ER structure. It is necessary to apply the same method to fibroblasts.
Figure 1.D is not a typical form to present ESC and iPSC. Please take a picture again. It is necessary to display the scale bar more accurately.
The content described in the line no. 457 is unnecessary. Please delete it.
Round 2
Reviewer 1 Report
-The main concern is a contradiction between the results that show more sensitivity to ER stress, which appears to cause toxicity in hPSCs and the conclusion of the authors, as indicated in the Abstract that “These results suggest a mechanism that possibly protects cells from stress in early embryogenesis”.
Upon initial stages of differentiation, protein synthesis is high and this is a natural cause of ER stress. The authors show that hPSCs have a lower capacity to adapt to ER stress and therefore instead of survival mechanisms, an apoptotic program is triggered, which is contrary to what happens in embryogenesis. This contradiction must be solved experimentally or interpreted correctly for the manuscript to be acceptable.
Response: Thank you for your constructive comments regarding the purpose and interpretation of our research. The function and abundance of organelles in each cell differs because of the unique functional roles of different cell types. Particularly, organelles mature in function and structure as needed for each cell type when the early embryo differentiates into individual cells. For example, cells that differentiate into pancreatic cells have a developed ER structure to produce secretory insulin protein. Such differences in the ER structure lead to diverse susceptibilities to ER stress. As the reviewer pointed out, in the early stages of development, protein synthesis is highly dynamic, resulting in the accumulation of misfolded proteins and maintenance of homeostasis. Early embryos have a response mechanism, the UPR, to protect cells from detrimental ER stress. Induction of the UPR appears to be essential for relieving and recovering from stress to allow for normal embryonic development (Refs). This study aimed to confirm that PSCs have different ER structures and ER stress susceptibilities as compared to fibroblasts. We agree that certain conclusions in the abstract are ambiguous because of the word choice. Sentences that were pointed out by the reviewer have been revised to improve clarity:
(line 33-34) These results suggest a mechanism that possibly protects cells from stress in early embryogenesis,
--> These results suggest that a mechanism exists which reacts to excessive stress during early embryogenesis,
Reviewer's response:
The main concern is still unsolved. The meaning of the new sentence at the end of the abstract is unclear “These results suggest that a mechanism exists which reacts to excessive stress during early embryogenesis, which can contribute to more effective cell therapy for disease.”
The sentence at the end of the Conclusions “Our results suggest a mechanism that protects cells from cellular stress in early embryogenesis.” is inconsistent with the results.
A discussion is needed about the consequences of the increased sensitivity of hPSCs to ER stress. Perhaps this is for many hPSCs to undergo apoptosis during development and for only a minority, with the correct microenvironment and signals to undergo adaptation and survive? Perhaps the TG treatment used in the experiments is much harsher than the normal conditions of ER stress caused by protein overload in the organism during development? These and other issues must be discussed for the manuscript to be acceptable.
-L.397-403. “Particularly, the BiP protein expression, an important factor that initiates the UPR signaling pathway upon ER stress, was relatively lower in hPSCs. BiP is bound to PERK, IRE1, and ATF6 to inhibit signaling in normal conditions… …However, since hPSCs displayed immature ER structure and low BiP protein amounts, they had a relatively weaker ability to regulate ER stress upon exposure, which led to rapid increases in the expression of UPR genes.”
The ratio of the UPR sensors to BiP should be measured. Perhaps there is just a smaller ER, therefore less BiP, but also less PERK, IRE1, and ATF6, so the ratio would be the same.
Response: We appreciate these insightful and constructive comments on our manuscript. According to the reviewer’s suggestion, we performed additional experiments to determine the ratio of BiP to the UPR sensors. We observed higher levels of IRE1a and ATF6a proteins in hPSCs than in fibroblasts, suggesting that the ratio of BiP to these sensors in hPSCs is much lower than in fibroblasts. Increasing evidence has suggested that increased BiP expression protects cells from ER stress-mediated cell death via decreased UPR induction (J Clin Invest. 2009 May 1; 119(5): 1201–1215, Diabetologia. 2013 May;56(5):1057-67. doi: 10.1007/s00125-013-2855-7). In contrast, reduced BiP levels make the cells more susceptible to ER stress with increased UPR induction (Cell Death & Differentiation volume 25, pages 2181–2194(2018), Archives of Biochemistry and Biophysics, 15 Sep 2007, 468(1):1-14). Based on the results of the current study and previously published observations, lower BiP levels and a lower ratio of BiP to UPR sensors in PSCs than in fibroblasts are closely related to the higher susceptibility of PSCs to ER stress (See below). This information has been added (line 414-420).
Reviewer's response:
Where is the new figure in the manuscript with the “additional experiments to determine the ratio of BiP to the UPR sensors” that you mention?
-L.404. Paragraph starting “Although the present study described the characteristics of hPSCs by inducing ER stress through TG treatment, these findings are useful to explain the cellular responses in hPSCs in the initial stages of differentiation.”
As explained above, during initial stages of differentiation there is high adaptation to protein load, contrary to what is seen in this study.
Response: We appreciate this suggestion and agree with the reviewer’s comments. We have made more specific changes in the texts as specified by the reviewer as follows: “The current study describes the characteristics of the response of hPSCs and induction of ER-stress via TG treatment, which may be useful for future studies of early human development.” (line 421–422)
Reviewer's response:
The new sentence is insufficient, there must be a thorough discussion of the implications of the present study, as mentioned above
